

# Towards quantitative viromics for both double-stranded and single-stranded DNA viruses

Simon Roux[1], Natalie E. Solonenko[1], Vinh T. Dang[2], Bonnie T. Poulos[3], Sarah M. Schwenck[3], Dawn B. Goldsmith[4], Maureen L. Coleman[5], Mya Breitbart[4] and Matthew B. Sullivan[1,6]

[1] Department of Microbiology, The Ohio State University, Columbus, OH, United States
[2] Department of Microbiology, Ha Long University, Uong Bi, Quang Ninh, Vietnam
[3] Department of Ecology and Evolutionary Biology, University of Arizona, Tucson, AZ, United States
[4] College of Marine Science, University of South Florida, St. Petersburg, FL, United States
[5] Department of the Geophysical Sciences, University of Chicago, Chicago, IL, United States
[6] Department of Civil, Environmental and Geodetic Engineering, The Ohio State University, Columbus, OH, United States

## ABSTRACT

**Background**. Viruses strongly influence microbial population dynamics and ecosystem functions. However, our ability to quantitatively evaluate those viral impacts is limited to the few cultivated viruses and double-stranded DNA (dsDNA) viral genomes captured in quantitative viral metagenomes (viromes). This leaves the ecology of non-dsDNA viruses nearly unknown, including single-stranded DNA (ssDNA) viruses that have been frequently observed in viromes, but not quantified due to amplification biases in sequencing library preparations (Multiple Displacement Amplification, Linker Amplification or Tagmentation).

**Methods**. Here we designed mock viral communities including both ssDNA and dsDNA viruses to evaluate the capability of a sequencing library preparation approach including an Adaptase step prior to Linker Amplification for quantitative amplification of both dsDNA and ssDNA templates. We then surveyed aquatic samples to provide first estimates of the abundance of ssDNA viruses.

**Results**. Mock community experiments confirmed the biased nature of existing library preparation methods for ssDNA templates (either largely enriched or selected against) and showed that the protocol using Adaptase plus Linker Amplification yielded viromes that were ±1.8-fold quantitative for ssDNA and dsDNA viruses. Application of this protocol to community virus DNA from three freshwater and three marine samples revealed that ssDNA viruses as a whole represent only a minor fraction (<5%) of DNA virus communities, though individual ssDNA genomes, both eukaryote-infecting Circular Rep-Encoding Single-Stranded DNA (CRESS-DNA) viruses and bacteriophages from the *Microviridae* family, can be among the most abundant viral genomes in a sample.

**Discussion**. Together these findings provide empirical data for a new virome library preparation protocol, and a first estimate of ssDNA virus abundance in aquatic systems.

Corresponding author
Matthew B. Sullivan,
mbsulli@gmail.com

## INTRODUCTION

It is now increasingly clear that microorganisms play a central role in all of Earth's ecosystems and processes. In every biome—from the human gut to the oceans, soils, and extreme environments that challenge life to succeed—microbes drive the nutrient and energy transformations that fuel these ecosystems (*Falkowski, Fenchel & Delong, 2008*; *Sommer & Bäckhed, 2013*). Microbial diversity was first revealed through universal marker genes studies (*Pace, 1997*), and has now advanced to genome-level characterizations helping revise our understanding of the microbial tree of life (*Rinke et al., 2013*; *Hug et al., 2016*), as well as elucidate the ecological and evolutionary roles of lesser-studied microbial lineages (*Wrighton et al., 2012*; *Castelle et al., 2013*; *Brown et al., 2015*).

Recent technical and theoretical advances are now also revealing that these microbial roles are modulated by co-occurring and co-evolving viruses (*Weitz, 2015*; *O'Malley, 2016*). Viruses are the most abundant biological entities on Earth, and have profound impacts on their microbial hosts through mortality, horizontal gene transfer and metabolic reprogramming (*Fuhrman, 1999*; *Suttle, 2007*; *Rohwer & Thurber, 2009*). Since most microbes and viruses remain uncultivated and viruses do not harbor a universal marker gene, community-level surveys of viruses typically rely on laboratory culture or reference-independent methods such as viral metagenomics (a.k.a. viromics). These studies have provided a view of viral diversity that complements knowledge obtained from isolate collections, and revealed new viral groups, evolutionary patterns, and virus-host interactions in multiple systems (*Edwards & Rohwer, 2005*; *Mokili, Rohwer & Dutilh, 2012*; *Fancello, Raoult & Desnues, 2012*; *Brum & Sullivan, 2015*).

Because natural community samples typically yield limiting input DNA, multiple displacement amplification (MDA) or whole genome amplification (WGA) are commonly used prior to sequencing library preparation in viromics studies (*Edwards & Rohwer, 2005*; *Brum & Sullivan, 2015*). While these MDA viromes routinely uncover new viruses (*Angly et al., 2006*; *Angly et al., 2009*; *Wegley et al., 2007*; *Kim et al., 2008*; *Willner et al., 2009*; *Ng et al., 2009*; *Rosario et al., 2009*; *Rosario, Duffy & Breitbart, 2009*; *López-Bueno et al., 2009*; *Roux et al., 2012a*; *Roux et al., 2012b*; *Labonté & Suttle, 2013a*; *Labonté & Suttle, 2013b*; *Zawar-Reza et al., 2014*; *Quaiser et al., 2015*; *Dayaram et al., 2016*; *Male et al., 2016*; *Steel et al., 2016*), the MDA step selects for small circular ssDNA templates, and unevenly amplifies linear genome fragments even when pooling independent reactions (*Yilmaz, Allgaier & Hugenholtz, 2010*; *Kim & Bae, 2011*; *Marine et al., 2014*). The alternative linker amplification (LA) or tagmentation (TAG) methods are quantitative ($\pm$1.5-fold) for dsDNA viruses, even from low input samples (100 femtograms, *Duhaime et al., 2012*) but strongly select against ssDNA templates (*Kim & Bae, 2011*). This leaves researchers to choose between quantitatively studying dsDNA viruses alone or pursuing questions constrained to discovery rather than ecology if interested in both ssDNA and dsDNA viruses.

The recently available Swift Biosciences 1S Plus kit for preparing sequencing libraries incorporates an adaptase step prior to linker ligation and amplification, which makes it efficient for both ssDNA and dsDNA templates (*Kurihara et al., 2014*; *Aigrain, Gu & Quail, 2016*). Here we use replicated metagenomic experiments to evaluate this protocol, hereafter

Table 1 Characteristics of phage genomes included in the mock communities.

| Genome type | Phage | Family | Host | Genome length (bp) | GC% | Theoretical proportion in MCA (low ssDNA) | Theoretical proportion in MCB (high ssDNA) | NCBI genome Id |
|---|---|---|---|---|---|---|---|---|
| dsDNA | PSA-HM1 | *Myoviridae* | PSA | 129,396 | 35.71% | 9.82% | 3.51% | KF302034 |
| dsDNA | PSA-HP1 | *Podoviridae* | PSA | 45,035 | 44.69% | 9.82% | 3.51% | KF302037 |
| dsDNA | PSA-HS1 | *Siphoviridae* | PSA | 36,769 | 40.50% | 9.82% | 3.51% | KF302033 |
| dsDNA | PSA-HS2 | *Siphoviridae* | PSA | 37,728 | 40.21% | 9.82% | 3.51% | KF302036 |
| dsDNA | PSA-HS6 | *Siphoviridae* | PSA | 35,328 | 44.78% | 9.82% | 3.51% | KF302035 |
| dsDNA | Cba phi38:1 | *Podoviridae* | Cba | 72,534 | 38.05% | 9.82% | 3.51% | NC_021796 |
| dsDNA | Cba phi18:3 | *Podoviridae* | Cba | 71,443 | 32.86% | 9.82% | 3.51% | NC_021794 |
| dsDNA | Cba phi38:2 | *Myoviridae* | Cba | 54,012 | 33.17% | 9.82% | 3.51% | KC821629 |
| dsDNA | Cba phi13:1 | *Siphoviridae* | Cba | 76,666 | 30.23% | 9.82% | 3.51% | KC821625 |
| dsDNA | Cba phi18:1 | *Siphoviridae* | Cba | 39,189 | 36.29% | 9.82% | 3.51% | NC_021790 |
| ssDNA | phix174 | *Microviridae* | E. coli | 5,386 | 44.80% | 0.92% | 32.47% | NC_001422 |
| ssDNA | alpha3 | *Microviridae* | E. coli | 6,087 | 44.56% | 0.92% | 32.47% | NC_001330 |

Notes.
PSA, *Pseudoalteromonas*; Cba, *Cellulophaga baltica*; E. coli, *Escherichia coli*.

named A-LA for Adaptase-Linker Amplification, alongside two existing protocols (MDA and TAG) for their ability to quantitatively amplify ssDNA and dsDNA viruses from two mock viral communities. Then, we apply the methods to aquatic samples known to harbor ssDNA viruses and estimate the relative abundance of ssDNA viruses.

## MATERIAL & METHODS
### Mock community generation
The ten dsDNA phages included in the mock communities were grown on *Pseudoalteromonas* or *Cellulophaga baltica* (Table 1, *Duhaime et al., 2011*; *Holmfeldt et al., 2013*). These were selected to represent the three main families of dsDNA bacteriophages (*Myoviridae*, *Podoviridae*, and *Siphoviridae*), a range of genome length (35–130 kb) and GC% (30–45%). The two ssDNA phages included were phiX174 and alpha 3, representing two distinct clades in the well-characterized *Microvirus* genus (from the *Microviridae* family), both grown on *Escherichia coli* (*Rokyta et al., 2006*).

Two mock communities were designed (A and B) corresponding to two contrasting situations with either low abundance of ssDNA viruses (MCA, total ssDNA ~2% of community) or high abundance of ssDNA viruses (MCB, total ssDNA ~66% of community).

Each virus to be included in the mock community was grown on its specific host, and viral capsids were obtained from lysates. The concentration of viral capsids was determined through SYBR Gold counting (*Noble, 2001*), using the wet-mound method (*Cunningham et al., 2015*), and two mixes of viral capsids corresponding to the desired relative proportion of viruses were created (MCA and MCB, Table 1). Although epifluorescence enumeration of some ssDNA phages can be challenging (*Holmfeldt et al., 2012*), SYBR-stained micrographs from phiX174 and alpha 3 were readily countable (Fig. S1).
For each mix, DNA was extracted with the QIAamp DNA Mini Kit (Qiagen 51304). Triplicate viromes were generated using DNA extracted from these two mock communities with three different sequencing library protocols (A-LA, TAG and MDA). This experimental design allowed us to evaluate the potential influence of both the DNA extraction step (same bias across all methods, as the same pool of DNA was used as input for all methods in MCA and MCB samples) and the DNA amplification step (different biases between methods) at three different levels: (i) the relative proportion of ssDNA vs dsDNA viruses, (ii) the relative proportion of individual genomes within ssDNA and dsDNA virus communities, and (iii) the coverage variation within a genome. The MDA library was generated using the GE HealthCare GenomiPhi v2 DNA Amplification Kit followed by NexteraXT DNA Library Preparation Kit, the TAG library using the standard NexteraXT DNA Library Preparation Kit, and the A-LA library with the Swift 1S Plus DNA Library Kit for Illumina. All samples were sequenced on the Illumina MiSeq platform.

## Mock community viromes: read quality control, assembly, and annotation

Raw reads were curated with Trimmomatic to remove adaptors, trim reads as soon as the base-calling quality dropped below 20 on a 4 bp sliding window, and remove reads shorter than 50 bp (*Bolger, Lohse & Usadel, 2014*). Trimmed and filtered reads from mock communities were mapped to the 12 reference genomes with Bowtie 2 (--non-deterministic option, default options otherwise, *Langmead & Salzberg, 2012*), and the normalized coverage of each genome (i.e., number of base pairs mapped at 100% identity to the genome normalized by the genome length and total number of base pairs sequenced in the virome) was used as a proxy for the relative abundance of each viral genome. Using a normalized coverage (i.e., number of reads mapped per position) rather than the total number of mapped reads per genome for estimating the relative abundance of each virus means that these relative abundance values did not have to be corrected for the different genome sizes. The expected number of viral genomes was calculated from the number of viral particles from each virus added in each mix, taking into account the fact that dsDNA viruses would provide twice as many genome copies as ssDNA viruses per particle following the first denaturation step of library preparation, and accounting for the low extraction efficiency of dsDNA genomes from lysates with the QIAamp DNA Mini Kit (estimated at 27% of DNA successfully recovered for the dsDNA viral genomes in these mixes; no similar bias was observed for ssDNA viral genomes). This DNA extraction efficiency was calculated based on the ratio between expected total DNA concentration (based on SYBR counts and the known genome size of the virus) and the observed DNA concentration (measured with PicoGreen) for PSA-HM1 (Table S1). Hence, the expected relative abundance of viruses in MCA and MCB viromes (Table 1) are calculated based on the expected number of genomes in the mix normalized by this dsDNA extraction bias. To verify if complete and accurate genomes could be reconstructed de novo from the mock community virome reads, the QC'd reads were assembled with Spades 3.6.2 using options "sc" and "careful," default options otherwise (*Bankevich et al., 2012*), and contigs ≥500 bp were compared to reference genomes with Nucmer (*Delcher, Salzberg & Phillippy, 2003*).

Within dsDNA viruses, the influence of mock community (A or B), library preparation method (MDA, TAG, A-LA), and genome on relative abundance (which should theoretically be 10% for every genome) was investigated with Kruskal–Wallis tests. Each genome's relative abundance was compared between A-LA and TAG libraries using a Wilcoxon two-sided test (Fig. S2). Similarly, statistically different ranges of coverage variations for each genome between the different library preparation methods were assessed with Wilcoxon two-sided tests (Fig. S3). All plots and tests were conducted with the R software (R Core Team, 2016) and the ggplot2 module (Wickham, 2016).

## Environmental virome sampling and processing

For freshwater lakes, three integrated water column samples were taken in April 2013 in Lake Superior (SU08M), Lake Erie (ER15M), and Lake Michigan (MI41M). Samples from three different depths (a "surface" sample at ~2 m for all lakes, an "intermediate" sample at 30 m for Lake Erie, 100 m for Lake Michigan and Superior, and a "deep" sample at 53 m for Lake Erie, 249 m for Lake Michigan, and 282 m for Lake Superior) were combined, since the lakes were not stratified at the time of the sampling. For each lake, 33 to 45L of water were 0.22 μm-filtered and viruses were concentrated from the filtrate using iron chloride flocculation (John et al., 2011) followed by storage at 4 °C. One seawater sample originates from the Tara Oceans expedition collection (sample T102S) and was processed as previously described (Pesant et al., 2015). Briefly, 20 L of seawater were 0.22 μm-filtered, and viruses were concentrated from the filtrate using iron chloride flocculation (John et al., 2011) followed by storage at 4 °C. In both cases, after resuspension in ascorbic-EDTA buffer (0.1 M EDTA, 0.2 M Mg, 0.2 M ascorbic acid, pH 6.0), viral particles were concentrated using Amicon Ultra 100 kDa centrifugal devices (Millipore), treated with DNase I (100 U/mL) followed by the addition of 0.1 M EDTA and 0.1 M EGTA to halt enzyme activity, and extracted with the QIAamp DNA Mini Kit (Qiagen 51304). The two remaining water samples were collected from 0 m and 100 m depths at the Bermuda Atlantic Time-series Study site in March 2011, where approximately 180L of seawater were concentrated using a 100kDa tangential flow filter, 0.22 μm-filtered, PEG precipitated, cesium chloride purified, and DNA was extracted using formamide (Goldsmith et al., 2015). All samples were sequenced on an Illumina MiSeq platform at the University of Arizona Genetics Core.

## Environmental viromes read quality control, assembly, and identification of viral contigs

For freshwater and seawater samples, trimmed and filtered reads (generated as for the mock community datasets, see above) for all libraries (MDA, TAG, and A-LA) were pooled for each sample and assembled with Spades 3.6.2 with the "sc" and "careful" options (Bankevich, Nurk & Antipov, 2012). All contigs >500 bp and with at least one complete gene were retained (representing on average 75% of trimmed and filtered reads, Table S5), and mined for contaminating cellular sequences. Contigs ≥5 kb were run through VirSorter (Roux et al., 2015) in the "virome decontamination" mode, all contigs not detected as viral were excluded from the final datasets, and prophage predictions were manually curated to distinguish cellular sequences from erroneous predictions (i.e., viral sequences wrongly

identified as a prophage). Another pipeline was applied to identify smaller viral contigs (<5 kb), which can be missed by VirSorter according to simulations (*Roux et al., 2015*): sequences with no significant BLAST hit (bit score > 50 and $e$-value < $10^{-3}$) against RefSeqVirus (i.e., no viral gene) and one significant hit (score > 50) against PFAM (i.e., one "known" gene) were considered as cellular and thus excluded (with the exception of "viral" PFAM domain, i.e., PFAM domains with the keyword "viral," "phage," "capsid," "virion," "terminase," "tail," or "portal"). This allowed us to keep in the dataset both sequences similar to known viruses, and sequences entirely new (i.e., all uncharacterized genes), which in a virome are most likely viral.

### Annotation of viral contigs from environmental viromes

QC'd reads from individual libraries were then mapped back to the contigs with Bowtie 2 (- -non-deterministic option, default options otherwise, (*Langmead & Salzberg, 2012*)) to evaluate the relative abundance of each sequence with each preparation method: contigs were considered detected in a library when ≥50% of the contig was covered, and the contig relative abundance was calculated from the contig average coverage normalized by the number of bp sequenced in the library. Contig affiliation was based on best BLAST hit against RefSeqVirus (thresholds: bit score > 50 and $e$-value < $10^{-3}$). Contigs with best BLAST hits to only ssDNA or dsDNA viruses were considered ssDNA or dsDNA viruses respectively, while the genome type of contigs with no hits or mixed affiliations (i.e., hits to both ssDNA and dsDNA reference genomes) was predicted based on their coverage in the different libraries: contigs detected in TAG libraries were considered dsDNA, while contigs only detected in MDA and/or A-LA were predicted as ssDNA. In order to take into account the fact that dsDNA genomes will provide twice as many templates than ssDNA genomes per single virus in A-LA and MDA viromes (because the first step of the protocol is dsDNA denaturation), the coverage of all affiliated and predicted dsDNA contigs was divided by 2, so that the relative proportion of contigs are approaching the relative proportion of viral particles in the sample.

### Scripts and datasets availability

All scripts and datasets used in this study are available on iVirus (CyVerse, http://mirrors.iplantcollaborative.org/browse/iplant/home/shared/iVirus/DNA_Viromes_library_comparison), as well as https://bitbucket.org/MAVERICLab/dna_viromes_library_comparison (for scripts).

## RESULTS & DISCUSSION

### Mock community benchmarking for ssDNA and dsDNA genomic amplification

Two mock communities containing a minority (MCA) or majority (MCB) of ssDNA viruses were established from 2 ssDNA and 10 dsDNA viruses (Supplemental Information, Table 1). DNA was extracted from each mock community, used as source material for constructing replicate sequencing libraries using MDA, TAG and A-LA methods, and sequenced to create viromes with approximately 1,000-fold coverage for abundant viruses (Table S2).

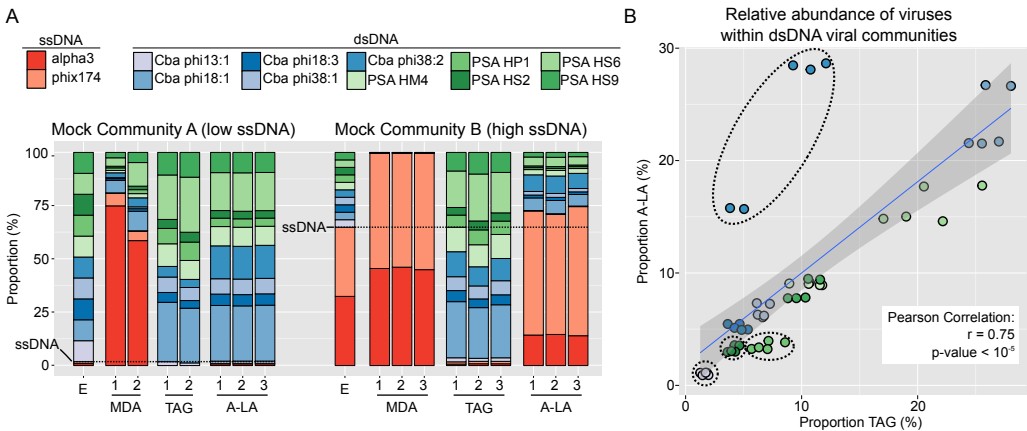

**Figure 1** **Comparison of amplification efficiency for ssDNA and dsDNA genomes of Multiple Displacement Amplification (MDA), Tagmentation (TAG) and Adaptase-Linker Amplification (A-LA) from mock community samples.** (A) Composition of mock communities' viromes prepared with MDA, TAG, and A-LA. For each community, the first bar displays the expected proportion of each virus ("E"), and replicates are noted with a number (1–3) when available. Expected proportions of ssDNA viruses are highlighted with a dashed horizontal line (1.8% and 64.9% of MCA and MCB respectively). (B) Correlation between the relative abundance of individual dsDNA viruses (within the dsDNA viral community) in TAG ($x$-axis) and A-LA ($y$-axis) viromes. The color code of circles is the same as in (A). Genomes for which the relative abundance distributions are significantly different in TAG vs A-LA are highlighted with dotted outline (Fig. S2).

As qualitatively observed previously (*Kim & Bae, 2011*), and here quantified, MDA systematically favored ssDNA viruses (∼30- to 40-fold), whereas TAG systematically selected against them (∼30- to 100-fold, Fig. 1A, Table S2). In contrast, A-LA correctly recovered the proportion of ssDNA viruses when they were in the majority (MCB, 1.1-fold variation), and slightly underestimated ssDNA viruses when they were in the minority (MCA, 1.8-fold variation, Fig. 1A, Table S2). These observations for all treatments were repeatable across duplicate or triplicate viromes (Fig. 1A). For A-LA viromes, the relative abundance of individual ssDNA viruses (within the ssDNA pool) was consistent across replicates, although not across the two mock communities (Table S2). This consistency across replicates suggests that the quantitative amplification of ssDNA viral communities through A-LA viromes is reproducible.

We next examined the variation in relative abundance estimates for the ten individual dsDNA viruses within the dsDNA pool in the mock communities, which revealed two main findings. First, the relative abundances of dsDNA viruses within the mock community were more variable than expected: each individual virus should represent 10% of the total dsDNA virus coverage, while observed relative abundances ranged from 0 to 30% (Fig. S2). Specifically, genome, and not sample or method, was the only significant factor explaining differences in relative abundance in a multi-factorial analysis (Kruskal–Wallis test, $p$-value < 2.2e–16, Fig. S2). The variation in relative abundances of each genome could be due to inaccurate viral particle counts and/or variable DNA extraction efficiencies for the input viruses. Notably, however, these relative abundance deviations are minimal
(10% ± 7–10, average ± st. dev.) compared to the many-fold variation typically tolerable in viral ecological counts (*Cunningham et al., 2015*). Moreover, these per-genome relative abundance estimates were minimally impacted by the choice of library preparation method: for each individual genome, the relative abundances were not significantly different (Wilcoxon test, *p*-value > 0.01, effect size < 0.8) for six of the 10 genomes between TAG and A-LA (Fig. S2, Table S2). This suggests that the current method used for dsDNA viruses (TAG) and the method evaluated here (A-LA) provide a relatively similar view of dsDNA viral communities (Fig. 1B). A notable exception was *Cellulophaga* phage phi38:2, for which relative abundance was systematically higher (2- to 3-fold) in A-LA than TAG samples (Fig. 1B). This genome did not have unusual size or GC content compared to the others (Table 1), so the mechanism for this deviation remains unclear.

Second, coverage variation along each genome indicated that MDA coverage was significantly more variable than A-LA and TAG for all genomes but one, and TAG more variable than A-LA for 6 of 10 genomes (Wilcoxon test, *p*-value > 0.01, effect size < 0.8), with highly variable coverage in TAG datasets for low GC genomes (Fig. S3, Table S3). Thus, among the tested methods, the A-LA protocol produces the most even coverage across dsDNA viral genomes.

In summary, these mock community findings suggest that A-LA was uniquely able to quantitatively recover ssDNA virus relative abundances from both mock communities, and also more quantitatively represented the coverage within dsDNA genomes. This indicates that A-LA would be the library preparation method of choice when targeting both ssDNA and dsDNA viruses in surveys of natural communities.

## Estimating the contribution of ssDNA viruses to aquatic viral communities

Given promising mock community benchmarking results, we next sought to apply these methods to establish their performances on natural communities, and to obtain first estimates of ssDNA virus sequence abundance in nature. To this end, we generated viromes for three freshwater and three seawater samples using the same library preparation protocols as above (MDA, TAG, A-LA; Supplemental Information). Overall, ssDNA viruses were detected in all samples, although these amounted only to 33–370 contigs in any given sample as compared to 14,000–99,000 dsDNA contigs (Table S4). However, because aquatic viruses are vastly under-represented in databases, a large proportion (35–71%) of the assembled contigs could not be confidently affiliated to either ssDNA or dsDNA viruses. Hence, we chose to generate a less stringent estimation of ssDNA contigs by adding all contigs not detected in TAG libraries (12,134–53,950 contigs, Table S4) to these BLAST-affiliated ssDNA sequences. Our reasoning is that unknown contigs detected in MDA or A-LA libraries (which will include ssDNA templates) and not in TAG libraries (strongly biased against ssDNA templates) likely originate from ssDNA genomes.

Based on A-LA viromes, which mock community experiments suggested were the most quantitative, the relative abundance (estimated through read coverage) of ssDNA contigs (conservatively identified by best BLAST hit to ssDNA virus genomes) was 0.03–4.68% and

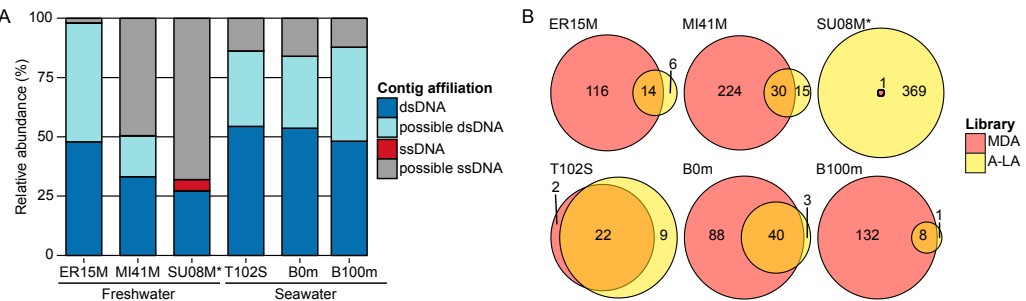

**Figure 2  Relative abundance of ssDNA vs dsDNA viruses in freshwater and seawater samples and estimated diversity of ssDNA viruses.** (A) Composition of A-LA viromes from 6 aquatic samples (based on the read coverage of assembled contigs). Contigs were affiliated based on best BLAST hit to NCBI RefSeq Virus ("dsDNA" and "ssDNA" contigs) or if not possible (no significant hit or mixed hits to both dsDNA and ssDNA genomes), based on their coverage in the TAG virome ("predicted dsDNA" if covered in TAG virome, "predicted ssDNA" otherwise). Relative abundance was calculated based on the coverage of each contig by virome reads. ER15M: Lake Erie, MI41M: Lake Michigan, SU08M: Lake Superior, T102S: surface sample of station Tara Ocean 102, B0m and B100m: surface and 100 m-deep samples from the Bermuda Atlantic Time-series Study site collected in March 2011. (B) Comparison of ssDNA viruses contigs recovered in MDA and A-LA library. For each sample, a Venn diagram depicts the number of contigs affiliated to ssDNA detected in MDA, A-LA, and both libraries (contigs detection based on a mapping of the library reads). *For sample SU08M, a limited number of quality-controlled reads were available for MDA and LA libraries (~1 order of magnitude less than for other samples).

0.005–0.03% in freshwater and seawater viral communities, respectively (Fig. 2A, Table S5). Meanwhile, the putative new ssDNA viruses (i.e., A-LA/MDA-only unknown contigs) could account for as much as 1.91–68.00% of freshwater and 12.15–15.98% of seawater DNA viral communities (Fig. 2A). Because this class of contigs might also include rare dsDNA viruses that would be haphazardly represented and not detected due to chance in the TAG libraries, these values of ssDNA abundance should be treated as lower and upper bounds. In addition, these upper bounds are likely over-estimations, especially in samples where few TAG reads are available, such as SU08M from Lake Superior (total ssDNA fraction estimated at 72.68%). Nevertheless, these still suggest that ssDNA viruses are less abundant than dsDNA viruses in four of six aquatic samples tested here (Fig. 2A, Table S5) although further work is required to address the recovery efficiency of ssDNA vs dsDNA viruses using various concentration and DNA extraction methods, as well as compare their decay rate and stability under different storage conditions, since both could influence the relative abundance of ssDNA vs dsDNA viruses in viromes.

Consistent with the mock community experiments, ssDNA viral genomes were systematically over-represented >10-fold in MDA viromes and under-represented >10-fold in TAG viromes, relative to A-LA (Table S5). This impacts rank-abundance distributions such that identifiable ssDNA viruses rank among the 10 most abundant contigs in MDA viromes, but are much lower ranked (~1,000–35,000th most abundant viruses) in A-LA viromes, and near or below detection limits in TAG viromes (Table S5). The only exception is sample SU08M, where ssDNA viruses rank as high as the 19th most abundant contig and have 16 additional viruses in the 100 most abundant viral sequences in the A-LA virome.

These abundant ssDNA viruses included bacteriophages (from the *Microviridae* family) and eukaryotic circular Rep-encoding ssDNA (CRESS-DNA) viruses (*Rosario et al., 2012*) (Table S6). Thus, even when ssDNA viruses as a whole do not represent a large part of the DNA viral community (affiliated ssDNA viruses account for only 3.68% of the reads in this sample), individual ssDNA viruses can be abundant.

The MDA bias towards enrichment for ssDNA viruses can be a positive attribute: MDA libraries captured 2–15 times more ssDNA viral genomes ("affiliated" ssDNA) than A-LA in four out of six samples (Fig. 2B). The two samples where MDA captured fewer ssDNA viruses represent unique situations: sample T102S had very few ssDNA viruses in any of its viromes, and the MDA library for sample SU08M was smaller by an order of magnitude relative to A-LA library due to multiplexing issues in the sequencing run (Table S5). Thus, when ssDNA viruses were available in the samples, and where sequencing depth was relatively consistent across library prep methods, MDA remains the clear method of choice to maximally enrich for ssDNA viruses if quantitative comparisons are not needed.

## CONCLUSION

The description of a large unsuspected genetic diversity of ssDNA viruses across multiple ecosystems (*Ge et al., 2011*; *Kim et al., 2011*; *Rosario, Duffy & Breitbart, 2012*; *Labonté & Suttle, 2013b*; *Eaglesham & Hewson, 2013*; *Quaiser et al., 2015*; *Dayaram et al., 2016*), and unique evolutionary patterns including gene exchanges between RNA and DNA genomes (*Krupovic, Ravantti & Bamford, 2009*; *Diemer & Stedman, 2012*), have highlighted ssDNA viruses as one of the most intriguing viral groups in viral ecology. So far, two main types of ssDNA viruses have been frequently detected in viromes: eukaryote-infecting CRESS-DNA viruses and bacteriophages from the *Microviridae* family. Novel and unusual ssDNA viruses continue to be isolated, particularly from eukaryotic and archaeal hosts (*Tomaru et al., 2012*; *Mochizuki et al., 2012*; *Kimura & Tomaru, 2015*). However, the relative abundance of ssDNA viruses among DNA viral communities remains an open and challenging question to address. Here, the use of A-LA library preparation protocol enabled us to quantify (±1.8-fold) the fraction of ssDNA and dsDNA viruses in natural communities. This revealed that ssDNA viruses are consistently present, but outnumbered by dsDNA viruses in all six aquatic samples tested. Nevertheless, individual ssDNA viruses were occasionally abundant, even when dsDNA viruses dominated the community. Hence, combined with the fact that ssDNA viruses likely infect a broad host range distinct from those of dsDNA viruses, the former should not be overlooked when investigating whole environmental viral communities and their impact on ecosystems.

## ACKNOWLEDGEMENTS

Seawater samples were available thanks to the *Tara* Oceans expedition and BATS cruises. We thank Dr. Bentley Fane from University of Arizona for providing phiX174 and alpha3 strains. We acknowledge support from the Ohio Supercomputer Center, which provided high performance computing and storage resources.

### Funding

This research was funded by the National Science Foundation (grant #1536989), the Gordon and Betty Moore Foundation (grants #3790, #GBMF2631), and the Flinn Foundation to MBS. SR was partially supported by the University of Arizona Technology and Research Initiative Fund through the Water, Environmental and Energy Solutions Initiative and the Ecosystem Genomics Institute. MB and DBG were supported by NSF grants MCB-0701984 and DEB-1555854. The funders had no role in study design, data collection and analysis, decision to publish, or preparation of the manuscript.

### Grant Disclosures

The following grant information was disclosed by the authors:
National Science Foundation: #1536989.
Gordon and Betty Moore Foundation: #3790, #GBMF2631.
Flinn Foundation.
University of Arizona Technology and Research Initiative Fund.
NSF: MCB-0701984, DEB-1555854.

### Competing Interests

The authors declare there are no competing interests.

### Author Contributions

- Simon Roux and Matthew B. Sullivan conceived and designed the experiments, analyzed the data, wrote the paper, prepared figures and/or tables, reviewed drafts of the paper.
- Natalie E. Solonenko conceived and designed the experiments, performed the experiments, analyzed the data, wrote the paper, prepared figures and/or tables, reviewed drafts of the paper.
- Vinh T. Dang analyzed the data, wrote the paper, prepared figures and/or tables, reviewed drafts of the paper.
- Bonnie T. Poulos and Sarah M. Schwenck performed the experiments.
- Dawn B. Goldsmith, Maureen L. Coleman and Mya Breitbart conceived and designed the experiments, wrote the paper, prepared figures and/or tables, reviewed drafts of the paper.

### DNA Deposition

The following information was supplied regarding the deposition of DNA sequences:
All sequencing data and scripts used in the study are available at http://mirrors. iplantcollaborative.org/browse/iplant/home/shared/iVirus/DNA_Viromes_library_ comparison.

### Data Availability

Bitbucket: DNA_viromes_library_comparison.
https://bitbucket.org/MAVERICLab/dna_viromes_library_comparison.

## Supplemental Information

Supplemental information for this article can be found online at http://dx.doi.org/10.7717/peerj.2777#supplemental-information.

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
