# Peer review of "Towards quantitative viromics for both double-stranded and single-stranded DNA viruses"

_PeerJ, doi:10.7717/peerj.2777_

## Round 0.1 · original submission · Minor Revisions

· Academic Editor

Minor Revisions

Please address each of the concerns and questions raised by the two reviewers. Importantly, please tone down the language, as not to oversell the protocol (kit) as new method and please address concerns about using appropriate and original citations. Please also provide more information on the quantification of phiX174 and alpha 3 to address concerns about the accuracy and utility of SYBR Gold for ssDNA phages. Lastly, please move Table S1 to the main article.

We very much look forward to a revised version.

Reviewer 1 ·

Basic reporting

Appropriate. Would like to see Table S1 moved to main article.

Experimental design

Appropriate. Would have preferred greater variety amongst the mock communities, but the design is adequate.

Validity of the findings

Appropriate.

Additional comments

Roux et al present their study examining the effect of different virome preparation methods on the relative abundances of viruses in the communities. They include mock communities with defined quantities of each virus and characterize the virome constituents using 3 different methods including MDA, TAG, and ALA methods, with the ALA method showing the least coverage bias particularly in regards to single stranded DNA viruses. I find this study to be useful and informative with regards to how certain biases in virome characterization may be limited by utilizing different preparation methods. I don't have any major critiques. My minor critiques are as follows:
1. Was there any correction for genome size in the authors calculations? Particularly as they use some longer siphoviruses, they should comment on any corrections that were made.
2. How did the authors choose the viruses for their mock community? I think Table S1 should be included in the main text of the manuscript with more detailed explanations as to why they chose their viruses for their mock communities.
3. Why the ratio of 2 ssDNA viruses compared to 8 dsDNA viruses in the mock community?
4. I don't care much for the exclusion of contigs <5kb, but the authors have used this technique quite often. I worry that they limit diversity of their communities using this technique, particularly when it comes to shorter viruses that may not be completely sampled.

Reviewer 2 ·

Basic reporting

pass

Experimental design

pass

Validity of the findings

pass

Additional comments

The authors present data a comparison study of three library construction procedures used in NGS viral studies that focus on ssDNA and dsDNA viruses and bacteriophage. While this is a nice comparative study, I do find that in many cases the authors reach to speculation to extend their data into an article when perhaps more is needed before it is ready for publication, i.e., can they try harder using existing methods for identifying the unknown pool – i.e., using sequence signatures or other ‘guilt by association’ methods. Also, this statement comes from the many times it is stated that more work is needed to truly understand the biases – Lines 229-242; 265-271; 277-280.

Major comments:
1) This is not a new protocol – as it is a kit that has been on the market for some time – we have been using this kit in the lab for over 1year to isolate ssDNA from environmental samples. While I appreciate the comparison and data provided about the protocol as it has been published before see Aigrain et al 2016, BMC Genomics (http://bmcgenomics.biomedcentral.com/articles/10.1186/s12864-016-2757-4) - here the authors compared multiple methods for DNA library construction only. Therefore it is more the implementation of protocol for this particular study.
a. I see the authors list Kurihara 2014 as a reference – I could only find a poster for this reference and there was no journal information in the reference list, please revise.
2) The generation of mock communities is a difficult task – even more difficult for phage/viruses – so I commend the authors for attempting this. However, I have concern about using SYBR Gold (a nucleic acid stain) for the accurate measurements of the ssDNA phage as it was previously published that this can be difficult (Holmfeldt 2012), which the corresponding author here, M. Sullivan, was also a part of this work. Could the authors provide evidence that accurate measurements for phiX174 and alpha 3 were obtained (even a previous report where this method was used would be good).
a. I see the SYBR Gold counting was referenced as Cunningham et al 2015. This is a widely used stain and this reference is not the first. I have seen this often by this group where the references primarily come from their own lab – while I understand the need for one’s own citations – I find it somewhat pervasive in this groups previous publications. A more thoughtful approach to previous work and others contributions is a general comment needed throughout this article.

Other comments:

In the introduction, ‘results’ section, the authors state that <5% of the DNA virus communities are ssDNA viruses – however they can be the most abundant viruses and bacteriophages in the sample. So the authors are saying that the most abundant family/group is <5% of the total. I find this hard to believe, please clarify and provide more information/data to support this. I know annotated viral sequences are often woefully low – however two from freshwater lakes and one from seawater (likely a productive region) seems too low. Also, no depths were given for the samples taken.

Line 129 – “… accounting for the low extraction efficiency…estimated at 27% of DNA successfully recovered for the ds DNA viral genomes …” Please provide data for how the extraction efficiency was measured or an appropriate reference for the estimation of 27%.

Line 245 – struggling, not the right tense and perhaps not the best word here as the method does not struggle (personification?).

Line 250 – based on the previous sentence about dsDNA coverage wouldn’t A-LA also be good for mixed communities of ss and ds – DNA?

Line 256 – “… first abundance estimates for ssDNA viruses in nature.” The authors need to revise to stress that this sequence abundance.

Line 271 – I think the back of the envelope calculations performed here are complete speculation as this is from the unknown pool of sequences. While the authors do try to say later that these should be ‘treated as lower and upper bounds’ I feel this should be removed or more carefully addressed.

---

## Round 0.2 · accepted · Accept

· Academic Editor

Accept

The reviewers concerns were addressed thoroughly and appropriately and we are excited about your contribution to PeerJ.

Reviewer 1 ·

Basic reporting

Appropriate

Experimental design

Appropriate. Improved details in current version.

Validity of the findings

Appropriate.